

# Where are the avalanches? Rapid mapping of a large snow avalanche period with optical satellites

Yves Bühler[1*], Elisabeth D. Hafner[1*], Benjamin Zweifel[1], Matthias Zesiger[2], Holger Heisig[2]

[1]WSL Institute for Snow and Avalanche Research SLF, Davos Dorf, 7260, Switzerland
[2]Federal Office of Topography swisstopo, Wabern, 3084, Switzerland
[*]These authors contributed equally to this work.

*Correspondence to*: Yves Bühler (buehler@slf.ch)

**Abstract.**
Accurate and timely information on avalanche occurrence are key for avalanche warning, crisis management and avalanche documentation. Today such information is mainly available at isolated locations provided by observers in the field. The achieved reliability considering accuracy, completeness and reliability of the reported avalanche events is limited. In this study we present the spatial continuous mapping of a large avalanche period in January 2018 covering the majority of the
Swiss Alps (12'500 km$^2$).

We tested different satellite sensors available for rapid mapping during a first avalanche period. Based on these experiences, we tasked SPOT6/7 data for data acquisition to cover the second, much larger avalanche period. We manually mapped the outlines of 18'737 individual avalanche events, applying
image enhancement techniques to analyze regions in cast shadow as well as brightly illuminated ones. The resulting dataset of mapped avalanche outlines, having a unique completeness and reliability, is evaluated to produce maps of avalanche occurrence and avalanche size. We validated the mapping of the avalanche outlines using photographs acquired from helicopters just after the avalanche period.

This study demonstrates the applicability of optical, very high spatial resolution satellite data to map an exceptional avalanche period with very high completeness, accuracy and reliability over a large region. The generated avalanche data is of great value to validate avalanche bulletins, complete existing avalanche databases and for research applications by enabling meaningful statistics on important avalanche parameters.

## 1  Introduction

Information on the occurrence and runout of snow avalanches is a key parameter for the development of effective hazard mitigation approaches for settlements and traffic infrastructure (Rudolf-Miklau et al., 2014;Bühler et al., 2018). Evidence on the locations and dimensions of avalanches is applied in hazard zone mapping, for the evaluation of protection measures and for the validation and further development
of numerical avalanche simulation software such as SAMOS (Sampl and Zwinger, 2004) or RAMMS



(Christen et al., 2010). Therefore, the number, size and release depth of avalanches with accurate location information are most important. For avalanche warning, comprehensive information on avalanche activity is important for the evaluation of the avalanche bulletin, the European avalanche danger scale (Meister, 1994) and further developments of avalanche danger assessment tools such as the
matrix of the European Avalanche Warning Services (EAWS) (Müller et al., 2016) or the conceptual model of avalanche hazard (Statham et al., 2017). Even though such information is of very high value for different applications, it is not available today in a satisfactory completeness and quality.

Avalanche occurrences are usually only reported today if they cause an obstruction to public
infrastructure, damage to personal property or are witnessed by local observers. In Switzerland all avalanches reported to the WSL Institute for Snow and Avalanche Research SLF that involved people or caused damage to property are stored in a database (Techel et al., 2015). Avalanches artificially released in ski resorts are also well documented, mainly for avalanche danger estimation with nearest neighbor models (Gassner et al., 2000) for legal reasons. The existing avalanche inventories tend
therefore to be biased toward damaging events or those reported from accessible terrain, whereas avalanches remain notoriously under-reported over larger regions. Since the Alps in Switzerland are quite densely populated, regions without reported avalanches are smaller than for example in Norway, but they do exist. Consequently, even under weather conditions with good visibility, only a fraction of all avalanches is captured. All those avalanches reported to the SLF from avalanche observers, ski
resorts, rescue organizations or other individuals are weighted by size, number and release type and then added up to the dimensionless Avalanche Activity Index (Schweizer et al., 2003). This index enables the comparison of avalanche activity per day throughout the winter and between years.

Avalanche detection systems based on time-laps photography (van Herwijnen and Heierli, 2009),
infrasound (Thüring et al., 2015), seismic signals (van Herwijnen et al., 2016;Heck et al., 2019) or radars and optical cameras (Meier et al., 2018) provide information on a local scale. Avalanche radar systems are already operationally applied to automatically close roads and railways in Switzerland. Depending on the system setup, an indication of avalanche size could be derived as well but is not standard information.
Remote sensing can provide spatially continuous information on avalanche occurrences over large regions including areas where no observers can acquire data. Optical data from airplanes and satellites with very high spatial resolution (0.1 – 0.5 m) was successfully applied to automatically map avalanche debris under cloud free conditions (Bühler et al., 2009;Lato et al., 2012;Eckerstorfer et al.,
2016;Korzeniowska et al., 2017). But these datasets are only available for selected regions and are hard to get on short notice. Recently, Unmanned Aerial Systems (UAS) have been successfully applied to document single avalanche events but they are not able to cover large regions mainly due to legal restrictions (Bühler et al., 2017;Eckerstorfer et al., 2016). Radar satellites have the advantage of acquiring data despite clouds and without daylight. Therefore, radar data has also been applied to
generate avalanche maps (Eckerstorfer and Malnes, 2015;Vickers et al., 2016;Eckerstorfer et al., 2017;Wesselink et al., 2017). Due to the coarser spatial resolution (3 – 30 m) and limitations by the observation geometries (radar shadow and layover), considerable parts of mountain regions cannot be





covered. Furthermore, the reliability of detecting medium to small size avalanches and in particular dry snow avalanches is not yet satisfactory. But this reliability to detect a large part of all occurred avalanches is key for the statistical analysis of an avalanche period.

The aim of this investigation is to generate a record of avalanche occurrences for an avalanche period over a large region (12'500 km$^2$) with a very high reliability. This dataset can then be used to do meaningful statistical analysis of the avalanche period and to produce a nearly complete database of avalanche runouts, that can be applied to validate the avalanche bulletin and hazard maps. Such a dataset did not yet exist.

**2   Avalanche periods and data acquisition**

**2.1 Avalanche periods 2018**

The year 2018 started snow rich in Switzerland. From 8 to 10 January a large snowfall event (period I) was responsible for up to 2 m of new snow in the Upper valleys of Visp and the southern Simplon region. This snowfall event was restricted to the southwest of Switzerland. In combination with a
previous snowfall event, which ended the 5$^{th}$ of January, the avalanche danger scale rose to very high danger (level 5) in the valleys of Visp and the Simplon region. This was the first use of danger level 5 in Switzerland since 2008 where it was not confirmed in retrospect. So, this is the first-time level 5 was reached in reality since February 1999, for 19 years.

Between the 15$^{th}$ and the 19$^{th}$ of January it snowed again - 30 to 100 cm, locally up to 160 cm. The biggest amount of new snow was measured in the northern Valais, the Vaud Alps, Upper Valais and the eastern part of the Northern flank of the Alps. The snowfall was accompanied by strong winds mainly from western directions causing big amounts of snow to drift over large distances. As a result, massive accumulations of windblown snow formed in particular within slopes facing east.

The period from the 21 to 23 January (period II) marked the largest three-day avalanche period in Switzerland since the avalanche winter 1999 (SLF, 2000). All mentioned events caused the total snow depths to reach new records for that time of the year at different long-term measurement stations with measurement periods of up to 80 years (https://www.slf.ch/en/avalanche-bulletin-and-snow-
situation/measured-values/description-of-automated-stations.html, last access: 30 April 2019). The night before the 21 January marked the beginning of the second series of snowfall events that caused 80 cm of new snow to fall in 24 hours in some areas. During the event the snowfall line rose up to 2000 m and dropped quickly again. Just like during the first period, snowfall was accompanied by strong winds, blowing from North to West. Hence, the accumulations formed prior to this event continued to grow.
During the second period 60 to 150 cm of new snow fell above 2200 m. Altogether the series of snowfall events accounted for two to three meters of new snow in the Valais, at the Northern flank of the Alps and in Grisons from Arosa to Samnaun and even more in the Northern Lower Valais and the Glarus Alps. The remaining areas received between one and two meters of new snow with less snow





towards the south. These amounts of snow were unusual and appear with an annuality of 15 to 30 years depending on the region.

The snowfall combined with strong winds and the rise in snowfall line caused the SLF to forecast very high avalanche danger (level 5) over a large part of Switzerland during the second period (Figure 2). During three avalanche bulletins the SLF forecasted very high avalanche danger (level 5). Many, very large (size 4) and in parts extremely large avalanches (size 5) were released causing damage to forest, landscape or infrastructure but fortunately no loss of life.

## 2.2 Rapid satellite data acquisition period I (8 – 10 January 2018)

During the first week of January, the SLF decided at an internal exceptional avalanche phase meeting that large scale spatial information on avalanche occurrence would be very helpful. Therefore, the SLF asked the Swiss Federal Office for the Environment (FOEN) to trigger the Swiss rapid mapping chain, which is funded by the FOEN. They requested the Federal Office for Topography (swisstopo) to order high and very high spatial resolution satellite data over the area, where danger level 5 was predicted by the SLF. The decision on what data to acquire and over which specific locations was made in close collaboration with the SLF. As it was largely unknown which sensors work best for rapid avalanche mapping, a large range of satellites were tasked for different regions (Table 1). The coverage of the different satellite sensors is pictured in Figure 1.

The time taken from the decision to task the satellites until the data acquisition was around 16 hours. It took another 12 hours until SLF had the data on their screens. For the acquisition of the optical satellite data it was crucial, that a cloud-free window appears shortly after the avalanche period. This was the case from 6 to 7 January 2018. The products were ordered with orthorectification performed by the data provider. The motivation was to have fast access to products that are directly applicable for avalanche warning and avalanche documentation.

Table 1: Acquired satellite datasets for the first period.

| Satellite | Acquisition date / time [UTC] | Spatial resolution [m] | Spectral resolution [nm] | Covered area [km²] |
|---|---|---|---|---|
| **WorldView-4 (optical)** | 07 January 2018, 10:23<br>07 January 2018, 10:24 | 0.3 PAN<br>1.2 Multispectral | PAN: 450 - 800<br>Red: 655 - 690<br>Green: 510 - 580<br>Blue: 450 - 510<br>Near-IR: 780 - 920 | 107<br>107<br><br><br>Total: 214 |
| **Pléiades (optical)** | 06 January 2018, 11:07 | 0.5 PAN<br>2.0 Multispectral | PAN: 480-830 nm<br>Blue: 430-550 nm<br>Green: 490-610 nm<br>Red: 600-720 nm<br>Near-IR: 750-950 nm | 130<br>143<br><br><br>Total: 237 |
| **SPOT6/7 (optical)** | 06 January 2018, 09:52<br>06 January 2018, 10:58 | 1.5 PAN<br>6.0 Multispectral | Blue: (455 nm – 525 nm)<br>Green: (530 nm – 590 nm)<br>Red: 625 nm – 695 nm)<br>Near-IR: (760 nm – 890 nm) | 265<br>804<br>1751<br>Total: 2820 |
| **TerraSAR-X (radar)** | 06 January 2018, 17:00<br>08 January 2018, 05:44 | 1 SpotLight<br>3 StripMap | X-band, 8 – 12,4 Ghz | 2 * 116<br>2 * 1500 |



| | 09 January 2018, 05:27 | | | Total: 3232 |
|---|---|---|---|---|

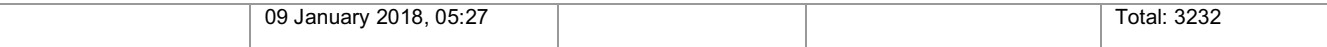

**Figure 1: Spatial coverage of the different satellite sensors for the two investigated avalanche periods. Hillshade © 2019 swisstopo (5 704 000 000), reproduced by permission of swisstopo (JA100118).**

## 2.3 Experience gained by analyzing the data from period I (8 – 10 January 2018)

The first optical datasets (Pléiades, WorldView-4, Spot7) were available on SLF screens less than 24h after the satellite tasking. This is very promising and demonstrates that the rapid mapping chain works well. The first radar datasets were also available after less than 24 hours after tasking. However, TerraSAR-X and can only acquire one dataset per overflight, limiting the coverable area within short time in particular for SpotLight mode acquisitions. Therefore, the next TerraSAR-X datasets were acquired on January 8 and 9. As snow avalanche deposits and in particular release zones can degrade quickly after avalanche release due to wind, new snow and melting, this is a drawback.

The optical datasets acquired over western Switzerland were nearly cloud free, the imagery acquired over central and eastern Switzerland had a cloud cover from 70 – 95%. The georeferencing performed



by the data provider of the very high-resolution imagery (Pléiades and WorldView-4) was clearly insufficient. Also, the orthorectification of the SPOT7 imagery was insufficient for avalanche documentation with large shifts and distortions. A main conclusion is, that the orthorectification has to be performed manually with an accurate digital elevation model (DEM). The SRTM applied by the data providers is insufficient in complex alpine topography. Consequently the datasets were oriented using bundle block adjustment and orthorectified by swisstopo based on its high quality DEM swissalti[3D] with a spatial resolution of 2 m, which is available for entire Switzerland (swisstopo, 2018). The results of this orthorectification were substantially better achieving a localization accuracy of better than 2 m in X and Y. Even though the spatial resolution of the SPOT6 (1.5 m) is coarser than the resolution of Pléiades and WorldView (0.5 m) the avalanche deposits and release zones are well visible. Due to the 12 bit radiometric resolution, also areas in cast shadow can be evaluated. A strong advantage of SPOT6/7 is that very large areas can be covered during one overpass of the satellite. Therefore SPOT6/7 is the platform of choice for future data acquisitions if a cloud free weather window is available close to the occurrence of the avalanche period. However, if a distinct and spatially limited hotspot (~250 km$^2$) of avalanche activity can be identified in advance, very high spatial resolution sensors such as Pléiades and WorldView could provide even more detailed information on avalanche release and deposit.

By analyzing the value of the TerraSAR-X data for rapid avalanche mapping, it was concluded that the interpretation of the data is too demanding on the short term and avalanche events are not clearly visible without advanced data preprocessing. More research is needed to evaluate possible filtering and imagery enhancements for this data. Furthermore, large regions within the steep avalanche terrain are not analyzable due to radar shadow and layover. Based on these findings TerraSAR-X data will not be ordered even though high spatial resolution radar data would be the only option for time periods where no cloud free weather window appears. However, the potential of radar data will be further investigated and compared to the results from Sentinel-1 data published by Eckerstorfer et al. (2017);Vickers et al. (2016);Wesselink et al. (2017);Eckerstorfer et al. (2016).

**2.4 Rapid satellite data acquisition period II (21 – 23 January 2018)**

Based on the conclusions drawn in section 2.3, it was decided to only acquire SPOT6/7 data for the avalanche period II as a cloud free weather period was predicted for 24 January 2018 over entire Switzerland. Swisstopo ordered a tasking of SPOT6 for 24 January 2018 on 23 January 4:35 p.m. (local time). The data was then acquired on 24 January 2018 at 11:04 a.m. or an area of approximately 12'500 km$^2$ across Switzerland and a part of Liechtenstein with a maximum extension of 300 km from the west to the east (Figure 2).

Based on the former experiences, orthorectifcation should be done manually e.g. by swisstopo. The product to be delivered by the data provider had therefore to be of type "Primary", with full radiometric resolution (12 bit) and pan-sharpened already. The data were oriented using bundle block adjustment within a photogrammetric block file. Besides automated Tie Point extraction, a number of 11 Ground



Control Points (GCP) was digitized manually. The achieved accuracy (RMSE) of the GCPs was of 1.23 meters in X, 0.83 meters in Y and 0.16 meters in Z.

The data was then delivered to SLF on February 6 in the early morning. SLF started the satellite data interpretation immediately. At the end of the avalanche period there was additional snowfall of about 0.3 m and strong winds that made avalanche detection in the optical imagery more difficult.

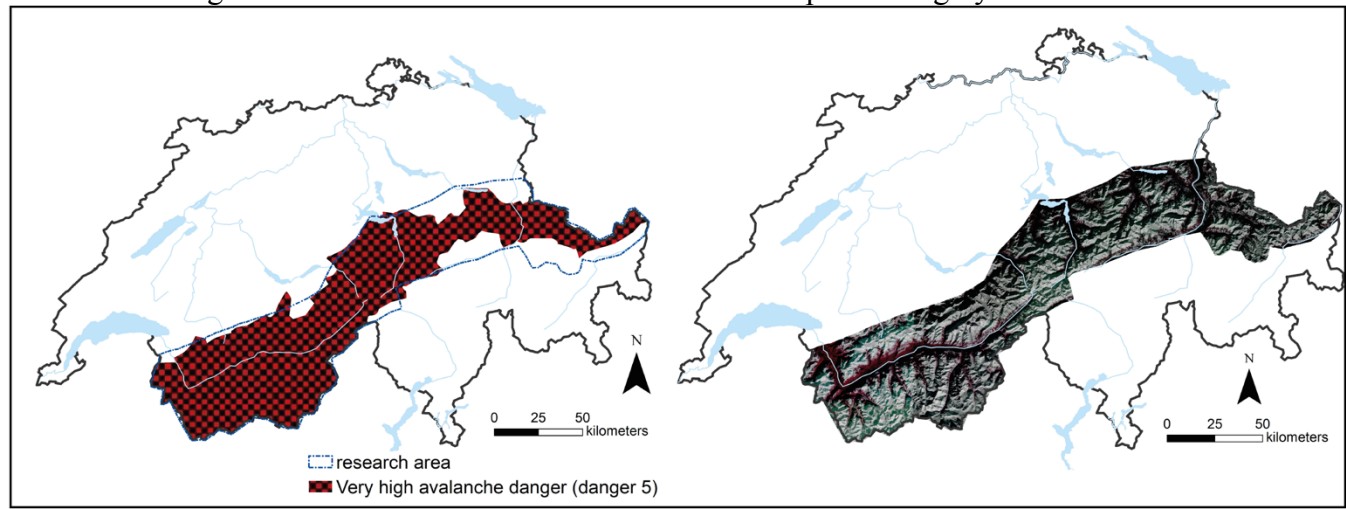

**Figure 2: study area and forecasted very high avalanche danger (danger 5) and overview on the acquired SPOT6 dataset (SPOT6 Data © Airbus DS 2018) acquired for period II.**

## 3    Mapping methodology

We used visual interpretation to identify and digitize avalanches as polygons over the whole study area. To improve visibility in both, illuminated and shaded areas, we modified contrast and brightness, used image stretching and gamma optimization. Since the optimal brightness and contrast differ for different illumination, we digitized the outlines in the sun and the shaded areas separately (Figure 3). Due to the lower reflectance of snow in the near infrared spectrum (Warren, 1982), the false color band combination NIR (green, red and near infrared bands) provides a clearly better visibility of the avalanches then the normal red, green blue (RGB) band combination. Additionally, the vegetation in forested areas and at the bottom of avalanche release zones is also better visible in the NIR combination than in RGB. Therefore, we only used NIR band combination for the mapping.

To perform a systematic mapping, the images were overlaid with a grid of 1,3 x 1,3 km. As additional information during interpretation, we used the 1:25 000 Swiss Map Raster 25 the summer orthophotos SWISSIMAGE 25 cm as well as the layer "Slope over 30 degrees" (all from swisstopo). The mapping itself was conducted using a scale of 1:5 000. After the initial mapping, the outlines were checked twice, first using the 1:25 000 map to check flow direction and second the satellite imagery for rechecking the accuracy of outlines and their completeness.



In order to keep all information that can be extracted from the images besides the outlines we defined an example key. The feasible attributes were described and an image example was added where appropriate. Part of the attributes like *quality of outline* or *avalanche type* need to be identified by the interpreter whereas *avalanche size* can be calculated later from the mapped polygons. The defined
5    attributes are listed in Table 2. The mapping itself was conducted in ArcGIS using "streaming" to map the outlines more easily as it allows tracing an outline without clicking every single vertex with the mouse. To remove the trembling effects of moving the mouse, a smoothing algorithm was applied in the end. To simplify the assignment of the attributes we build a geodatabase and implemented the characteristics as coded values.

**Table 2: Description of avalanche attributes**

| Attribute | Characteristics | Description |
|---|---|---|
| **Quality outline** | exact | The outline from starting to deposit zone is clearly visible as a whole. |
| | estimated | The outline from starting to deposit zone is clearly visible in most places. In-between where the outline cannot undoubtedly be identified the clearly visible parts are connected considering terrain. |
| | created | Only starting or deposit zone are visible. Considering terrain the rest of the outline is created. Same is true for avalanches "cut off" at the edge of the images. |
| **Avalanche type** | Slab | Slab avalanches start with an initial failure in a buried weak layer. When the weak layer is underneath a cohesive snow slab a crack can propagate. If the weak layer fractures extensively and the slope is sufficiently steep a slab avalanche will release. |
| | Loose snow | Loose snow avalanches start from a single point and enlarge pear-shaped. |
| | Glide snow | Glide snow avalanches form due to a loss of support between the snowpack and the smooth ground. |
| | unknown | Only the deposit zone is visible or the type can no longer be identified. |
| **Avalanche size** | | The European Avalanche Warning Services (EAWS) provide a standardized avalanche size classification depending on destructive potential, runout length and dimensions. The first two parameters cannot be extracted from aerial imagery; hence we used the classification relying on area only already in use in Protools, a tool for avalanche safety services to ease avalanche mapping (https://www.slf.ch/de/services-und-produkte/protools.html). |

| Size | Name | Size in m$^2$ |
|---|---|---|
| 1 | Small avalanche | 10 to 500 |
| 2 | Medium avalanche | 501 to 10'000 |
| 3 | Large avalanche | 10'001 to 80'000 |
| 4 | Very large avalanche | 80'001 to 500'000 |
| 5 | Extremely large avalanche | > 500'000 |

| Attribute | Characteristics | Description |
|---|---|---|
| **Aspect** | | Aspect is split up in the eight main aspects (N, NE, E, SE, S, SW, W, NW) and represents the mean aspect for each avalanche in the release zone. As release zones weren't resigned separately they were calculated using a threshold for the ratio of avalanche width and height difference. The areas meeting this criterion and being over 27° steep were used to calculate the aspect using a 10 x10m height model. For avalanches 230m$^2$ and smaller the aspect was calculated using the whole mapped area. The aspect was calculated for "created" avalanches as well but might only be used considering the implications of the quality of outline. |
| **Trigger type** | natural | All glide snow avalanches plus all avalanches outside of skiing areas and those in skiing areas where additional information on their spontaneous release was available. |





| | explosive | Avalanches where the explosion point was visible in the imagery and where information on the artificial release was available. | |
|---|---|---|---|
| | unknown | The rest of the avalanches in skiing areas where there was no additional information available. | |
| **Start Zone Altitude** | height of the highest point of the outline in m above sea level | Both are calculated using the outline after mapping. Those values are given for all avalanches but need to be used in light of the quality of outline. | |
| **Deposit Zone Altitude** | height of the lowest point of the outline in m above sea level | | |
| **Type fracture** | Old snow fracture | Practically only old snow fractures close to the ground may be identified as the ground will then be shining through red in the Infrared. All other fractures are unknown as they may not be differentiated in the imagery. | |
| | New snow fracture | | |
| | unknown | | |
| **fracture width** | The release zones calculated for the aspect were also used to determine the fracture width. The fracture width is defined as the length of the longest contour line (10m equidistance) in the release zone and is given in meters. For loose snow avalanches no fracture width is given as they per definition start from one single point. | | |
| **comment** | In the comment section information is given on suspected forest damage and potential damage to infrastructure. Additionally, "abnormalities" such as a very high mud content, release zone of the avalanche inside constructions for avalanche protection or similar things are recorded. The comment section is not limited in its content, basically all information that might be of importance to future users of the avalanche outlines is allowed. | | |

## 4 Results and discussion

In total we mapped 18'737 individual avalanche outlines with an area of 936 km$^2$ (Hafner and Bühler, 2019). This accounts for 7.5 % of the total investigation area covered by satellite imagery. A selection of examples showing different illumination conditions is depicted in Figure 3.





**Figure 3:** Examples of mapped avalanches in imagery optimized for brightly illuminated areas (1) and for regions with shaded areas (2). A deflection dam that worked well by changing the flow path of the avalanche is marked as red line in (1). (SPOT6 Data © Airbus DS 2018). Overlapping avalanches area is determined by interpretation of the flow direction from the topographic map and by differences in contrast between different depositions. The position of the avalanches is indicated by the numbers in the map on the top right.

The 18'737 avalanches are not equally distributed over the whole investigation area. Visualizing the avalanche area as shares of 1x1 km grid cells reveals where a high density of avalanche activity is present (Figure 4). In some regions, the high density had been suspected as observers had reported at least a fraction of the avalanches. In other regions there is hardly any human activity in January and therefore no avalanche activity information was available. Clearly visible in the map are also the effects of the high snowfall line (partly up to 2000m) – in the Rhône and Alpine Rhine Valley where there were no avalanches close to the valley bottom.

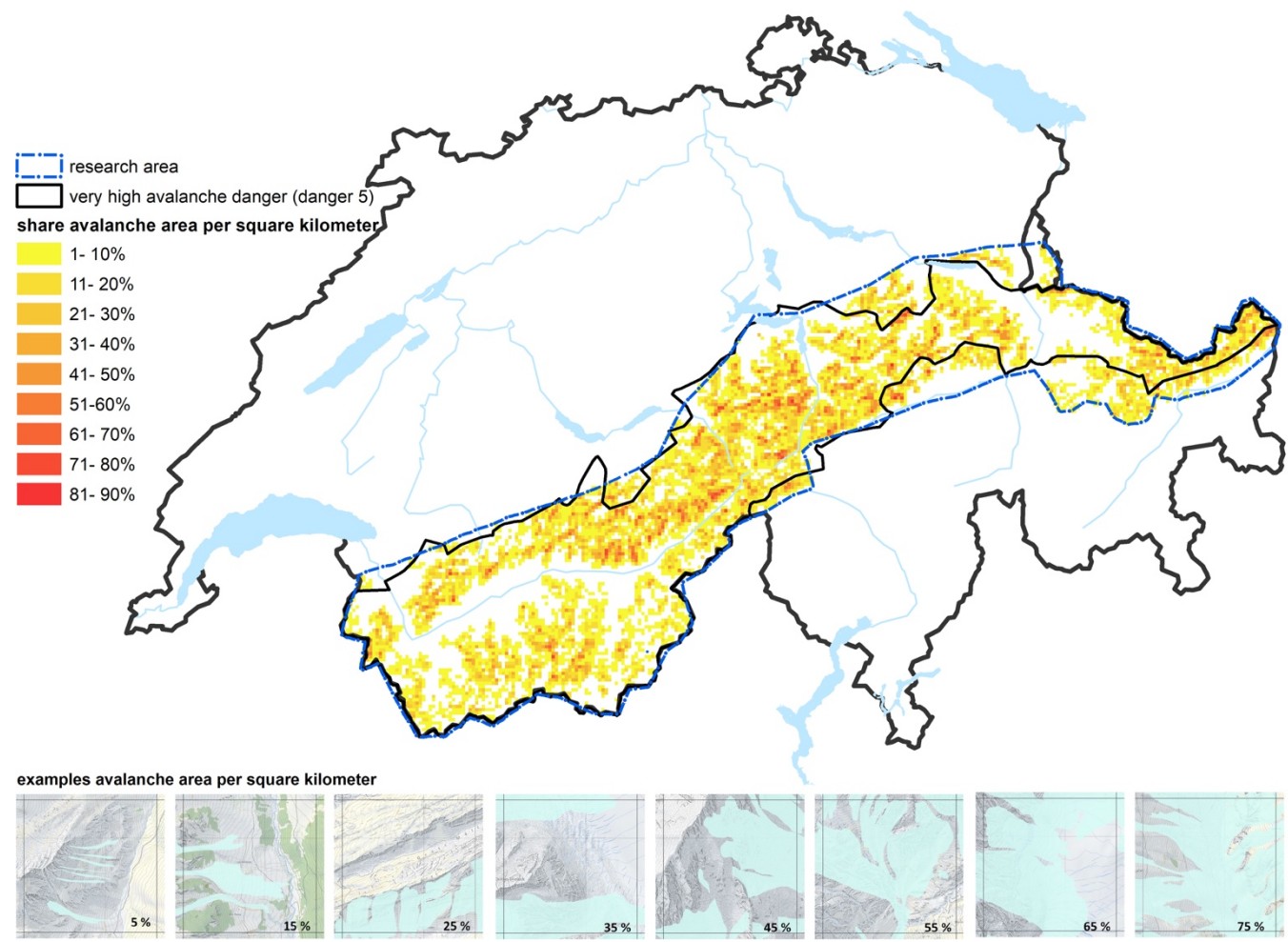

**Figure 4: Avalanche density per square kilometer. Pixmap © 2019 swisstopo (5 704 000 000), reproduced by permission of swisstopo (JA100118).**

5   33 % of the avalanches could be mapped with exact outlines, 58 % with partly estimated outlines and
for 9 % we had to create parts of the outline (Figure 5) by expert interpretation. The number of created
outlines is highest in the aspects North to West (Figure 5). In the opposite aspects we mapped the
highest percentage of exact outlines. With an approximate sun altitude of 15° and an azimuth of 141°
(see suncalc.org for 24 January, 10 am, Andermatt, CH) during data acquisition, most shaded areas in
10   the satellite images are expected opposite of the sun. These shaded areas coincide with the largest share
of created avalanches (Figure 5) caused by a poorer visibility in the shaded areas and a clearly better
visibility in illuminated areas. The estimated outlines don't show a strong variation with aspect
indicating the lack of such a correlation. The avalanches we could map with an exact quality outline



tend to be smaller than the ones that were estimated or created. Estimated and created outlines occur for avalanches of all sizes but more frequently within avalanches of bigger size.

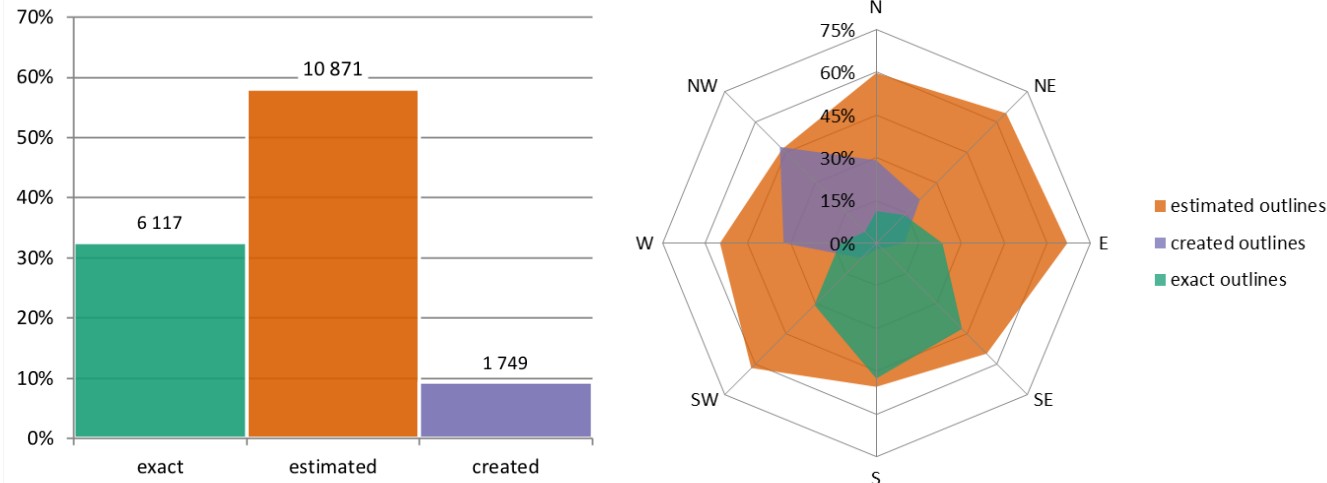

**Figure 5: amount and exposition of exact, estimated and created avalanche outlines**

With 72 %, the majority of avalanches mapped were slab avalanches (Figure 5). The share of glide
5   snow and loose snow avalanches is 11 % and 3 % respectively. For the rest of the avalanches, the type could not be identified either because the outline had to be created in the starting zone or wind, snowfall and melting between avalanche release and data acquisition had made the identification of avalanche type impossible.

10   Avalanche size is one of the attributes that can be calculated after mapping (Table 2). The by far biggest amount of avalanche are large size avalanches (size 3) followed by medium size avalanches (size 2, Figure 6). This is expected for very high avalanche danger levels. Additionally, multiple very large and extremely large avalanches (size 4 and 5) occurred.
91 % of size 1 (small) avalanches are glide snow avalanches which are very well visible in satellite
15   imagery because of the contrast between the exposed ground and the surrounding snow in particular within the near infrared band. The smallest slab avalanche we could identify had an area of 415 m$^2$ and is therefore a rather large size 1 (small) avalanche (Table 2). The spatial resolution of 1.5 m is at the limit to identify smaller (size 1) avalanches. Since those size 1 avalanches are of least interest in a large avalanche period, this limitation is of minor importance.





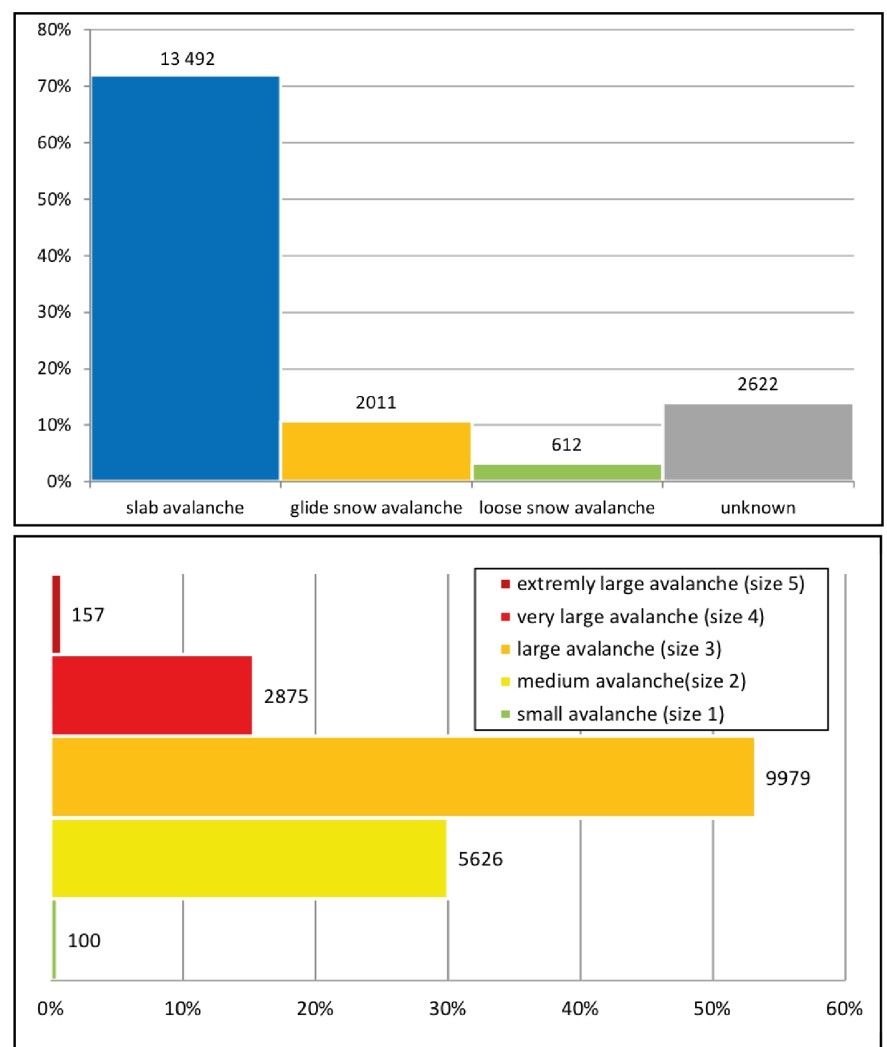

Figure 6: share of the different mapped avalanche types (top) and sizes (bottom)

## 4.1 Age of the mapped avalanches

While mapping, we found several avalanches that were already released prior to the second avalanche
5   period. In order to understand the dimensions of this problem and to estimate how many avalanches are
old ones, we used the SPOT 6/7 images taken for test purposes after avalanche period I (section 2.2).
We chose two test areas- the Mattertal und part of the Lower Engadine Valley- making up 5% of the
total research area.

10  In those test areas we investigate how long the avalanches remain visible. For the Mattertal we used
images from the 6, 12 and 24 January 2018 and in the Lower Engadine Valley from the 8 and 24
January 2018. For each point in time we classified the avalanches mapped from the 24 January imagery



whether they had been released before avalanche period II as shown in Table 3. In total, we classified the visibility of 550 different avalanches for all images taken before the 24 January.

**Table 3: avalanche classification according to age**

|  | description |
|---|---|
| **Yes** | The avalanche has been released and is visible with the same dimensions and deposition pattern. |
| **No** | No avalanche is visible/ has been released yet. |
| **Partly** | A part of the avalanche mapped from SPOT 6 imagery from the 24th of January is already visible (Figure 7). This is equally applicable for avalanches in the same avalanche track with a decisively different deposit pattern. |
| **Not visible** | As mentioned in chapter 2.2 the images from the first period have some cloud cover and some distortions from rectification that might make the particular area of interest invisible. |

In the Mattertal we found 22 % of the mapped avalanche had already released completely before the 12 January (Yes) and 12 % were even visible in the images from the 6 January (double Yes). In the Lower Engadine Valley we identified 20 % of the avalanches already in the image from the 8 January (Yes). Additionally, between 16 % (Lower Engadine Valley) and 25 % (Mattertal) have been partially released

before the second avalanche period (Partly).

These numbers may not be transferred directly to the whole dataset since both test sites have had very strong avalanche activity during the first period. This was not the case for the majority of the investigation area. Therefore, 25 % of old avalanches can be taken as an upper limit value. For the

whole dataset we estimate 10 – 20 % of the avalanches mapped originating before avalanche period II (21 - 23 January). Additionally, we found that large avalanches remain visible longer. The oldest avalanches (released 6 January or earlier) show the highest percentage of avalanches with an area larger than 10 000 m² (92 %). This number is decreasing to 78 % for the more recent avalanches from avalanche period II. This is expected for larger deposits as it takes more time for them to melt and more

snow to cover their rough surface structure.

We conclude that avalanches seem to be visible for a longer period of time on satellite imagery than we expected (especially after heavy snowfall like in our case). This is an important finding for future avalanche mapping campaigns. Firstly, because the fear of "missing" very large and extremely large

avalanches from the beginning of a several days long avalanche period seems to be unfounded. Secondly especially later in winter the high percentage of older avalanches visible might make it hard to differentiate different avalanche periods. In those cases, it would be valuable to have additional pre-event satellite data that maybe exist in satellite databases. In any case it makes sense to check pre-event imagery from continuously operation satellites such as Sentinel-2 or Landsat. Even though their spatial

resolutions are coarser (10 - 30 m) very large and extremely large avalanches could be identified.





Figure 7: SPOT 6 images from 12th and 24th of January from close to Oberrothorn, VS (SPOT6 Data © Airbus DS 2018). The purple avalanche outlines are those visible on the imagery acquired first (left). One of these avalanche outlines is still visible after the second avalanche period twelve days later (right). The blue outlines released during the second avalanche period.

## 4.2 validation approaches

For validation we were confronted with the difficulty of finding a meaningful dataset for such an extensive mapping campaign. At the SLF, avalanches in the region of Davos are mapped systematically over the whole winter from photographs taken in the field. Unfortunately, the quality of outlines generated with this technique only allows for a comparison of methods and not for a real validation. A comparison using very well visible avalanches from SPOT 6 imagery, showed that the satellite-based mapping is clearly more accurate than the manual mapping from ground-based photographs.





Therefore, we decided to use the WSL Monoplotting tool (Bozzini et al., 2013, 2012) to digitize validation imagery acquired from helicopters on 24 January 2018. This tool allows for georeferencing and orthorectifying of photographs in order to produce georeferenced vector data by drawing directly on the pictures and exporting the vector data for use in GIS-Systems. In order to georeference photographs

with the Monoplotting tool, Control Points (CPs) and a Digital Elevation Models (DEM) are needed. We applied the 2 m resolution swissalti[3D] from swisstopo, which has a nominal accuracy of 0.5 m below tree line (~ 2100 m a. s.l.) and 1- 3 m above tree line (swisstopo, 2018). Due to the lack of clearly identifiable features within avalanche terrain, we often had to use trees and rocks as CPs. This reduces the accuracy of rectification as they are hard to exactly identify in orthoimagery. Hence, we specified

that the maximum error had to be under 5 m and the mean error under 3 m for each picture we used for validation in order to keep the distortion through validation data as small as possible. Applying this restriction, 13 avalanches were mapped in full or in parts from suitable helicopter images. By overlaying these outlines with those mapped from the satellite images (outline quality exact or estimated only) we were able to calculate a trend in accuracy for the mapped avalanche area (Figure 8).

The overall accuracy for the mapped area is 73 % with an omission and commission error of 16 % and 11 % respectively (Table 4). In illuminated terrain, we achieved a decisively better overall accuracy (80 %) than in shaded areas (64 %). Especially the omission error, i.e. the avalanche area we missed is higher in shaded areas (25 %). This confirms our impression that the correct identification of avalanche outlines in shaded terrain is more difficult than in illuminated areas. This approach can only validate the

mapping accuracy for the avalanche surface area not for the number of avalanches we missed or we falsely mapped. No reliable ground truth dataset exists for such a validation.

**Table 4: Accuracy of the mapped avalanche area**

|  | total | illuminated areas | shaded areas |
|---|---|---|---|
| **correct classification** | 73% | 80% | 64% |
| **omission error** | 16% | 9% | 25% |
| **commission error** | 11% | 12% | 10% |





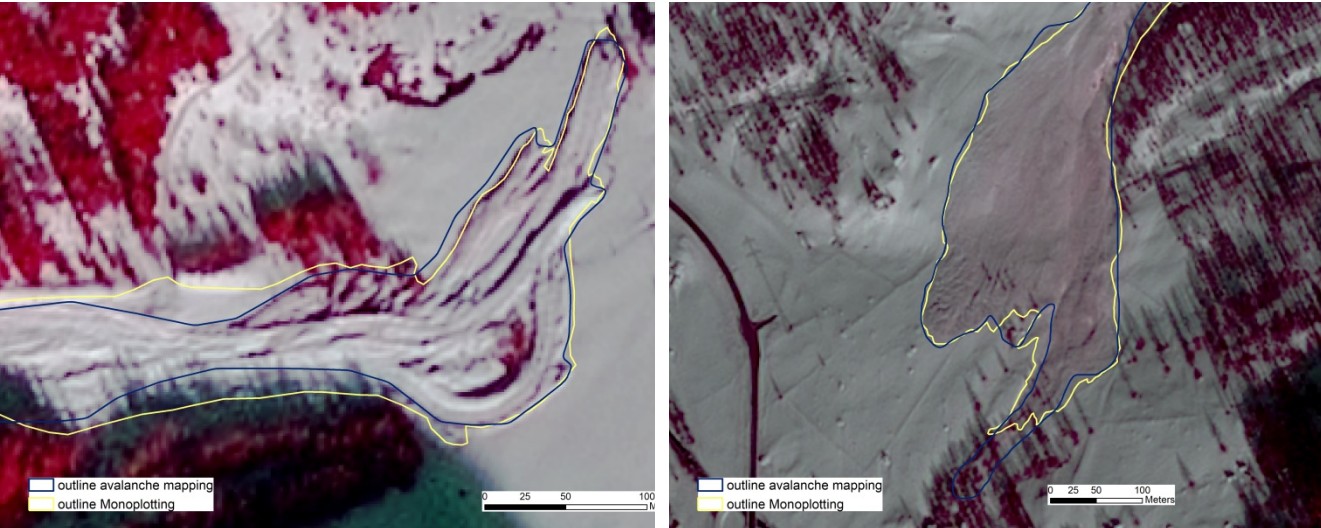

**Figure 8: comparison of mapped outlines (blue) with those from the Monoplotting tool (yellow) for two different avalanches. The left avalanche is located in Trient, VS and the one to the right in Simplon, VS. Especially in the shade, under trees and with thin deposit there are considerable differences. In both images the background is SPOT 6 imagery from the 24 January (SPOT6-Data © Airbus DS 2018).**

The following list of important problems and uncertainties were discovered during the mapping and
    validation process:

- Older and new avalanches in the same avalanche track are mapped as one single avalanche.
- Avalanches flowing through forest and over rock walls are hard to identify.
- Some avalanches which are completely in shaded areas might be "invisible" and can therefore
not be mapped.
- Avalanches which have been covered by new snow or have snow deposited on them are poorly
    visible.
- High surface roughness caused by wind may lead to wrong identification of avalanches.
- Cornices with snow that has rolled down beneath may be incorrectly classified as a slab
avalanche.

## 4.3 Potential improvements and follow up analysis

The manual mapping method we applied is very time consuming. Mapping the 18'737 avalanches over
the entire 12'500 km$^2$ took approximately 600 hours. 300 hours for the initial mapping and 300 hours
for checking and corrections. Therefore it would save a lot of time and costs if the data could be
analyzed automatically for example by machine learning (Zhang et al., 2016) or by the object based
    approaches applied to higher spatial resolution optical data (Bühler et al., 2009;Lato et al.,
    2012;Korzeniowska et al., 2017). We did not yet start to follow this track but we estimate a big potential
    for successfully detecting the 6'117 avalanches mapped with exact boundaries (33 %). For the





remaining 12'620 avalanches (67 %), however, we estimate a low success rate as it requires a lot of background knowledge and interpretation to map those avalanches as a whole outline (see section 3).

The mapped avalanches described in this paper are a unique training and validation dataset as it
contains a large number of individual avalanches in differing topography and illumination conditions over a very large region. The same applies for the validation of existing mapping product from radar satellite data (Eckerstorfer et al., 2017;Vickers et al., 2016;Wesselink et al., 2017).

To further improve the speed of the manual mapping we plan to confine the area that could theoretically
be covered by avalanches. Bühler et al. (2018) developed an algorithm for large scale hazard indication mapping combining automated release area delineation with numerical avalanche dynamic simulations with RAMMS (Christen et al., 2010). Such a mask could limit the area to investigate considerably saving time and costs. Conversely the generated avalanche dataset is a very good validation dataset for the newly developed large-scale hazard indication mapping processing chain (Bühler et al., 2018;Bühler
et al., 2013).

Another important application is the documentation of forest damages. In comparison with pre-event satellite data forest destruction can be mapped with the same methodology as for the avalanches. Such information is crucial for a fast and target-oriented management of protection forests (Bebi et al., 2009)
and the initiation of necessary complementation of protection infrastructure (Rudolf-Miklau et al., 2014). This is in particular useful in remote regions, hardly accessible in winter.

## 5  Conclusions and outlook

Based on the experience gained with satellite data acquired with different sensors in a first rapid mapping study in January 2018, we documented the following large avalanche period based on 1.5 m
resolution multispectral SPOT6 satellite imagery. Over an area of 12'500 km$^2$ in Switzerland we manually mapped the outlines of 18'737 individual avalanches. This number is surprisingly high and is to our knowledge the largest, most complete and most reliable documentation of a large avalanche period. Avalanches in shadows are mostly visible but harder to map exactly than avalanches in well illuminated areas.

Using parts of the imagery from the first rapid mapping study we found some avalanches to be visible even 18 days after they were first captured on an image. This was a surprising finding and is important as it proves that the largest and therefore most interesting avalanches are visible for quite a long time though with decreasing accuracy of the outline.



By using orthorectified photographs taken from helicopters one day after the acquisition of the SPOT6 satellite data, we can estimate the mapping accuracies of the mapped avalanche areas. We achieve an overall accuracy of 73 % with a clearly higher chance of missing avalanche parts in shaded areas (error
of omission 25 %). As the avalanche outlines mapped operationally in the region of Davos based on photographs taken from the ground are of clearly lower accuracy and completeness then the satellite mapping, we are not able to meaningfully validate the completeness of the mapping.

The applied data and methodology proof to be applicable to document large avalanche periods over
very large regions such as entire countries with a high reliability and completeness. Such data is valuable for various applications such as the validation of the avalanche bulletin, the validation of hazard mapping, the complement of avalanche cadasters and the validation of other avalanche mapping products. We plan to include the mapped avalanche outlines into the Swiss avalanche databases.

Based on the findings of this study, again SPOT6/7 data was requested for an area of 9'500 km$^2$ in eastern Switzerland on 16 January 2019 following an avalanche period with very high avalanche danger (level 5). The acquired data was once more cloud free, allowing for a mapping of avalanches. To better estimate the mapping quality, we acquired unmanned aerial system (UAS) photogrammetric data over four large avalanches in the region of Davos. This will allow us to make further remarks on the quality
of mapping. During this avalanche period in 2019, the snowfall line was much lower than in 2018 producing many large dry snow powder avalanches, causing significant forest destruction. As we now have SPOT6/7 data from January 2018 and 2019, the potential to accurately document this forest damages are high. The mapping 2019 and the follow-up studies will provide refined information on the potential of optical satellite data for large scale avalanche studies.

**Data availability**

The produced avalanche outlines described in this paper including a description of the attributes will be available on ENVIDAT ([www.envidat.ch](www.envidat.ch)) on the final publication of the revised paper *(Hafner and Bühler, 2019)*.

**Acknowledgments**

The authors thank the Federal Office for the Environment FOEN and the canton Valais for partially funding the satellite data. We also thank the cantons Valais, Graubünden, Bern, Obwalden and Uri as well as Liechtenstein for partially funding the data analysis. We thank the Swiss Airforce for providing helicopters to acquire reference imagery and Claudio Bozzini for support with the Monoplotting Tool. We thank the SLF avalanche warning service for valuable feedback and support.





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
