# Peer review of "Where are the avalanches? Rapid SPOT6 satellite data acquisition to map an extreme avalanche period over the Swiss Alps"

_The Cryosphere, 2019_

## Referee Comment (RC1) · Markus Eckerstorfer (Referee) · 24 Jul 2019

Mapping nearly 19000 avalanches in about 600 hours is quite a feat. As somebody that has mapped many thousands of avalanches myself (although in radar images) I am very impressed by the amount of work that went into creating this unique dataset. It should be also stated that nobody has every assigned this amount of avalanche activity to a single extreme event. We often state that avalanches are rare natural phenomena, however, I wonder if this statement holds true when applying new technology to an old and seemingly trivial problem: counting how many avalanches when and where release in a given area. The field of avalanche activity monitoring using satellite data is

young and exciting and this paper is a great contribution. I am kindly asking to consider the issues that I am raising and the questions that I have in the following:

Short summary of major issues and questions: 1. Reporting of period I: There are two sections with a table and a figure introducing a lot of different data that are not used in the study. I got very curious about a comparison of avalanche detection in these different data, however, this is not done. I think the authors could consider shortening this section substantially, or even deleting it. 2. Mapping methodology: I very much believe in visual interpretation of satellite images as a sound scientific method. However, not everyone does. It would be great if you could give much more detail on how the mapping was done. I have raised a couple of questions further below that deal with uncertainties in digitizing the outlines and assigning avalanche attributes to the detected features. 3. Validation approaches: Validating satellite image interpretations is very difficult, however, highly critical if we want to install more trust in our methods from other scientific communities. The accuracy of the mapped outlines is one thing, a simple error of omission study (are the mapped features really avalanches and what are you missing?) is another possibility that I would like you to consider including. 4. In your aim of this paper you are stating that a nearly complete database of avalanche activity could be used to validate the avalanche bulletin. This is, however, missing, and could easily be added in the discussion. 5. This is a case study from Switzerland. Since satellites cover pretty much all snow-covered mountain areas worldwide, it would be interesting to read a short discussion on the potential of doing this anywhere else. Here data availability, revisit time, max spatial coverage, cloud cover and shadow problematic and also acquisition costs could be discussed.

Here are my detailed comments:

Title: I believe that the title does not entirely represent the work that you have done. Data acquisition was rapid; however, mapping took 600 hours. Is 'large' the correct term when talking about a period of time? And in the end, you are only using Spot6/7 data, so maybe you could simply say so. Maybe the title could be something like: 'Mapping of an extreme avalanche cycle in the Swiss Alps using Spot6/7 optical images'.

1 Introduction You are introducing the concept of AAI in the introduction. Is that of interest here to the reader? If yes, could you calculate the AAI for period II and show it in results? Thereby you could compare period II to other extreme events (1999 winter for example) and arrive at a quantitative assessment of period IIs magnitude.

2 Avalanche periods and data acquisition 2.1 Avalanche periods 2018 There are a lot of Swiss place names in this section. I – and probably the majority of readers – are not familiar with where all those places are. You could either provide a map, or leave this place names out.

2.2 Rapid satellite data acquisition period I (8-10 January 2018) As mentioned above, I am a little bit unsure if all the detailed information you are providing here is necessary to get the full story of your paper. You are introducing a lot of data in a table and a figure, however, only a very tiny fraction of it is actually used for age-tracking of detected avalanches. At the same time, this section and also 2.3 read like discussion with some valuable information about which datasets to use for mapping purposes. So I am wondering if 2.2 and 2.3 could be deleted and some of the information could go into the discussion instead?

2.4 Rapid satellite data acquisition period II (21 – 23 January 2018) Figure 2 caption Study area in the caption, however research area in the legend. Also, maybe write 'danger level 5'.

You are writing that 30 cm of snow and strong winds made avalanche detection more difficult. This is a statement that kind of contradicts the tone of the rest of the paper, stating that you have great success in mapping avalanches in optical imagery and your dataset is reliable.

3 Mapping methodology You used additional information for the mapping. How were these data layers used? Did you mask out areas that were not considered avalanche

terrain or did you blend these layers in and out?

The flow direction check, was that done also by visual interpretation or with a GIS tool?

The rechecking of accuracy of outlines and completeness. Could you give more details here as to what is meant by accuracy and what completeness means?

Could you explain in some sentences if one or more experts visually interpreted the images? If the rechecking was done by the same person as the mapping? I think this is operational implications as to whether the person mapping was an expert in remote sensing image interpretation or if this is a task that a less experienced person with avalanche background could also do?

What is meant by feasible attributes that were described and which image examples were added when appropriate? These statements are not entirely clear to me.

Figure 2: I wonder if you could delete this figure since you are showing the extent of forecasting area and study area in Figure 4.

Table 2: Description of avalanche attributes This is a great list of attributes that is of critical importance when mapping avalanche activity. I wonder when in the process this list was established. Before all the digitizing took place or after? I am missing a differentiation between dry and wet snow avalanches. Since you state correctly that wet snow avalanches are easier to detect in radar images, I wonder if the same holds true for optical images? How did you attribute avalanche trigger and fracture type to the detections?

4 Results and discussion 7.5 % of the total investigation area (you have called it also research area and study area) were covered by avalanches. It might be probably more interesting how much of avalanche terrain was covered?

Figure 3: I am very interested in which attributes were assigned to these avalanches according to Table 2. What is the quality outline of these avalanches and could you maybe indicate by different line coloring where it is exact, estimated and created? I

am in particular also interested in where the start and deposit zone altitude are and the fracture width. In example 2), the digitized avalanches show only parts of the slide path and the depositional area. Why are you not showing the start zone? How sure are you of the interpretation of the two avalanches in the shadow and is there maybe a third avalanche in the path to the left? Can you explain a little bit more in detail how the flow direction interpretation works when dealing with overlapping avalanches and indicate it in the figure?

Figure 4 and related text: I am curious about in which regions high avalanche activity was suspected. You are also talking about the effects of a high snowline in two valleys; however, I don't know where these valleys exactly are. Could you provide more topographic information in Figure 4 for all non-Swiss readers? I wonder if a shaded relief as background would improve this figure too. Finally, the examples of avalanche density per km2 appear to be transparent. These small squares are very hard to see. Could you please improve the image quality or make these squares a little bit larger? In these examples you are going from 5 to 75 % coverage in 10 % steps. I wonder if you could do the same in the map. You would end up with fewer classes which would maybe increase the contrast between classes. Or how about using the AAI from Schweizer 2003 to represent avalanche activity?

You state that the avalanches with exact outline tended to be smaller than the others. Do you have any idea why that is?

The largest portion of avalanches falls under the estimated category. Is it possible for you to quantify how much of the outline had to be estimated and what the most common problems were that led to the estimation of the outline? I am also interested which parts of the avalanche had to be most often estimated and which avalanche type?

Figures 5 and 6: This is just a minor issue; however, these two figures could be visually slightly more pleasing. Related to Figure 6, I would find it very interesting to see

examples of the different avalanche types and avalanche sizes. If it is too much for this article, maybe in an appendix figure.

4.1 Age of the mapped avalanches Where are the Mattertal and Lower Engadine valley?

Can you estimate or even quantify how many avalanches were not visible anymore after the 6 or 8 January because wind or snowmelt removed them?

Figure 7: The largest of the blue outlines seems to go uphill (from the left upper corner towards the middle of the image) and then downhill again. Could you indicate flow directions maybe? These images are very hard to interpret as it is not really possible to understand where up and down is. Maybe contours or other topographical information would help here. Are these the smoothed lines or the raw digitized ones?

4.2 Validation approaches This is a good effort to validate your visual interpretations even though the test dataset is very small. Furthermore, I think that the results from Table 4 and Figure 8 are very good and quite satisfying. However, I am wondering why you are not using the dataset of field observed avalanches in the Davos area to at least analyze how many of these avalanches you were able to digitize? Digitizing the outline correctly is one problem, however, as you state in your list of important problems and uncertainties, there are possibilities that you miss avalanches entirely or that you interpret high surface roughness wrong. As somebody with little experience at interpreting optical images I am not fully convinced that I would have found the avalanches presented in Figures 3 and 7. I would like to suggest therefore to consider doing a two-part validation: Number one showing the reader that you are capable of detecting avalanches by comparing to a field observed dataset of avalanche activity and number two the validation you did so nicely already.

Figure 8: I would be interested in seeing a comparison for the starting zone as well since you are only showing the depositional parts. Furthermore, the two Spot 6 images as background appear to have different resolution, with the left one having a lower one.

In your list of problems and uncertainties you state that some old and new avalanches were mapped as single avalanches. Do you think these avalanches were given attribute 'Yes' or 'Partly' according to Table 3?

4.3 Potential improvements and follow up analysis You are talking about using machine learning for automatic analysis. Could you expend on these thoughts a little bit more and explain how such a workflow might look like and which kind of algorithms would be suitable here. You mention that a lot of background knowledge and map interpretation skills are involved in mapping. Is there a way that machine learning could incorporate these skills too? And how confident are you that a ML algorithm could classify avalanche types?

You state that you have a detailed avalanche terrain map for Switzerland. Wouldn't it be great for this study to run this map over all your detections and see if any of the avalanche outlines are outside the avalanche terrain and thus rather unlikely to be real avalanches (given they are not very large avalanches). One could at least report on how many features where outside and maybe even visually recheck if these features were avalanches. This would be another great validation tool!

Was danger level 5 confirmed by your analysis?

What do you think about the up and coming swarm satellites and their potential for rapid mapping?
* * *

---

## Author Comment (AC1) · 2 Aug 2019

Dear Markus

Thank you very much for your constructive and careful review of our paper. Please find in the following our answers to the issues and questions you raised:

1. Reporting period:

It is true that we only use SPOT6/7 optical satellite data for the mapping of the second avalanche period. However, it was extremely important to go through the process of tasking and interpreting the other satellite datasets. Only by comparing and interpreting

all other sensors we could come to the decision to further apply SPOT6/7. Therefore we see the chapters 2.1, 2.2 and 2.3 as essential parts for understanding the big picture. In particular in chapter 2.3 we name a lot of our findings that led to the decision to further use SPOT6/7 and that are of great interest for potential future applications. We believe this part is very interesting for the readers and we want to keep it to help them understand the scope of these investigations for our mapping. We did not further expand this part on our findings as it would be in our opinion too much detail for this paper.

2. Mapping methodology:

Thank you very much for the helpful comments you raised. We will carefully go through all the points and expand the information wherever possible in the revised manuscript to make the methodology even more clear.

3. Validation approach:

You are very right in stating that a validation of the number of missed and the number of falsely identified avalanches would be very meaningful information. We tried hard to set up a validation like this. Unfortunately, it is not possible. The reason is the quality of the manual mapped avalanche dataset of Davos for the time period needed. In the paper we already state that "For validation we were confronted with the difficulty of finding a meaningful dataset for such an extensive mapping campaign. At the SLF, avalanches in the region of Davos are mapped systematically over the whole winter from photographs taken in the field. Unfortunately, the quality of outlines generated with this technique only allows for a comparison of methods and not for a real validation. A comparison using very well visible avalanches from SPOT 6 imagery, showed that the satellite-based mapping is clearly more accurate than the manual mapping from ground-based photographs". In some cases it is not even clear, which of the avalanches clearly visible in the SPOT6 data corresponds the one mapped in the region by hand. In ongoing studies, focusing on the mapping from 2019 where we were able to collect

more supplementary data, we will further investigate different mapping methods and will present strengths, weaknesses and capabilities.

4. Validation of the avalanche bulletin:

In the meantime, we performed different validation approaches for the bulletin. We will add a paragraph with some examples to the revised manuscript.

5. Application in other regions:

We will add a section in the discussion to debate the potential of the presented methodology for other alpine regions.

Title:

You are right with your remark on the title. We will change it to "Where are the avalanches? Rapid SPOT6/7 data acquisition to map an extreme avalanche period in Switzerland" based on your suggestion.

AAI concept:

The AAI is mentioned because we are trying to list other methods used to capture avalanche activity. We could calculate the AAI for the time period where we expect most avalanche releases happened. Then we could compare the "normal" AAI added up for several days with our results We have tried that and as mentioned in the introduction, it shows that only a fraction of avalanches that are released are normally reported.

Local names:

We will drop the local names in the manuscript and replace them with globally understandable terms, e.g. southwestern valley etc. to make it better understandable for non-Swiss readers. If local names are used, we will depict them in a map or figure.

Figure 2:

We would like to keep this figure as it gives the readers a better resolved overview on

the entire research area.

Table 2, avalanche attributes:

The example key/ the attributes were defined before the mapping was conducted. The attributes heavily rely on the ones used in Protools (https://www.slf.ch/en/services-and-products/protools.html) as we wanted to ensure compatibility. Wet and dry snow avalanches may not be distinguished in the mapping. Per definition the classification wet or dry avalanche refers to snow temperature in the release area of the specific avalanche. If a dry avalanche entrains wet snow in the avalanche track it still remains a dry avalanche. To distinguish these fine differences is not possible in our experience. Additionally, rain after the avalanche release will increase contrast (and the snowfall line was at times very high in our case). For those reasons the humidity of the avalanches remains unknown for the whole dataset. But we have benefited from the better contrast of wet snow for identifying avalanches just like for the radar images you mentioned.

The attribute fracture type is taken from the images. As noted in Table 2, this is only possible for old snow fractures close to the ground (as the ground will then be shining through red in the near infrared band). All other fractures are unknown as they may not be differentiated in the imagery.

As also described in Table 2, the trigger type is partly taken from the images and partly from additional information. All glide snow avalanches are natural releases as they can't be artificially triggered. Avalanches where the points of explosion are visible have been artificially released. Additional information (i.e. Protools) is used to classify remaining artificially released avalanches and natural ones especially in ski areas. The remaining avalanches are classified as natural except they are inside a ski area then they will be classified as unknown. This procedure was designed especially for mapping in exceptional avalanche situations and may have to be adapted should triggering by skiers also be an option.

Avalanche terrain:
We are not yet there that we calculated all avalanche terrain for Switzerland. We are now calculating different regions in Switzerland, Italy and further abroad after the method of Bühler et al. 2018. So we cannot give this number now.

Figure 3:

We are showing only part of the outlines on purpose because the focus of the example to the right was to show the differences between illuminated and shaded areas. The outlines of the two avalanches in shade are estimated- the deposit and parts of the track can be identified well. Overlapping avalanche outlines occur when crowns of slabs are visible or the contrast between tracks/ deposits suggests two separate avalanches. Flow direction plays a minor role in that process. We will perform some test to find a way to display even more information without overloading the figure.

Figure 4 and related text:

High avalanche activity was suspected where very high avalanche danger (danger level 5) was forecasted. We already have tried a shaded relief as background and it did in our opinion not improve the map. So we will again try to add more topographic information without overloading the figure. As for the classes- a reduction of classes would not happen if the steps from the examples were used since the maximum value is 88%. But we could use steps of 15% to reduce classes if that is wanted, but in our opinion the 10% steps are the best option. AAI is already dealt with when mentioned the first time in this answer.

Avalanche outlines:

For smaller avalanches the probability that parts of the outline are in the shade or not so well visible is simply smaller than for the ones with longer outlines. We have no record about what part of the outline had to be estimated. But per definition estimated means that a couple of meters in-between can't be identified. This implies that part of deposition, track and release area are identifiable in an exact manner. The estimation

of an outline might be caused by wind blurring part of the "avalanche border", by part of the outline not being illuminated or passing under trees. The share of the different avalanche types for the quality of outline estimated is 85% for slab avalanches, 9% for unknown type, 4% for glide snow avalanches and the remaining 2% for loose snow avalanches. Concerning created avalanches, for 85% the deposit could be identified but the release area had to be created.

Figures 5 & 6:

Providing more examples would go beyond this papers scope but they are several additional examples provided within the example key that is delivered with the dataset that is available on request on Envidat (Hafner and Bühler 2019).

Age of the mapped avalanches:

The outlines shown in Figure 7 are the final and therefor smoothed outlines. We will try to improve figure 7 with additional topographic information (contour lines).

Figure 8:

The two SPOT examples have the same resolution but are shown at a different zoom level. The old and new avalanches being mapped as one avalanche laying inside the "age testing area" were certainly given the attribute partly at some point if on an image before the 24th of January the avalanche was partly already there. But as mentioned in the definition "partly" is equally applicable for avalanches in the same avalanche track with a decisively different deposit pattern.

Potential improvements and follow up analysis:

We will extend this section based on your suggestions. However We will not be able to give a detailed outline for machine learning strategies as we just started to research into this direction. We already state in the paper "We did not yet start to follow this track but we estimate a big potential for successfully detecting the 6'117 avalanches mapped with exact boundaries (33 %). For the remaining 12'620 avalanches (67 %),
however, we estimate a low success rate as it requires a lot of background knowledge and interpretation to map those avalanches as a whole outline (see section 3)" The point on the avalanche terrain map is already answered above.

In the revised version of the paper we will carefully consider all further points you raised and try to incorporate them into the document.

References:

Bühler, Y., von Rickenbach, D., Stoffel, A., Margreth, S., Stoffel, L., and Christen, M.: Automated snow avalanche release area delineation – validation of existing algorithms and proposition of a new object-based approach for large-scale hazard indication mapping, Nat. Hazards Earth Syst. Sci., 18, 3235-3251, 10.5194/nhess-18-3235-2018, 2018.

Hafner, E., and Bühler, Y.: SPOT6 Avalanche outlines 24 January 2018. SLF (Ed.), EnviDat, 2019.

---

## Referee Comment (RC2) · Anonymous Referee #2 · 5 Aug 2019

Avalanche hazard mapping is time consuming and requires avalanche phenomena expertise and a very good knowledge of the terrain. The purpose of this paper is to map avalanche deposits over large areas using high resolution satellite imagery. The authors examined two episodes of high avalanche activity in January 2018. The selected periods were associated with exceptional snowfall over entire Switzerland and with a very high level of avalanche danger scale. The authors manually detected a very large number of avalanches on SPOT6 / 7 images combined with accurate auxiliary information on the topography of the study area. They have shown the full potential of high-resolution optical imaging to help monitoring areas affected by avalanches. This is crucial for many applications including the identification of potentially hazardous areas

and to evaluate and / or validate physical models of avalanche hazard forecasts. The subject treated by this paper is therefore very important and the authors make a very good contribution to this field. The paper is easy to read, contains a lot of information (probably to be reduced) and provides all the necessary details for readers.

I would like the authors to consider some elements of improvement to increase the readability of the manuscript.

General and specific comments:

1- regarding the overall organization of the paper, I consider that there is an imbalance with the highly detailed section 2 relative to other parts of the paper. This section could be well shortened by targeting only the observations used in this study. 2- Figures: Maps are difficult to understand because there is a lack of information: administrative boundaries, elevation lines, names of some cities... 3- this avalanche mapping exercise is very interesting and I wonder how easily it can be replicated elsewhere? apart from financing issues, what could be the limits regarding image acquisition time, the preprocessing, the time needed to identify the avalanche signatures ? 4- As correctly stated by authors, this is a very useful database for many researchers working in this field. Do the authors intend to share it ? 5- Mapping methodology: it is unclear to me if the avalanche mapping was done entirely manually ? If the polygon detection method is completely visual, would some sort of classification based on the NIR band combination be used to pre-select avalanche zones ? 6- Mapped avalanches: I like Figure 4 which shows the density of the avalanche zones. I would be great to have another figure that completes it with some statistics about the altitudes min / max, orientations and the relative size of avalanches. color palettes can be improved. 7- The authors mentioned possible combined use with large scale hazard mapping (RAMMS simulations): could you give more details about it ? 8- Did the authors look at the mapping stats specifically according to the types of forests, the water areas? 9- Age of avalanches: very interesting section. It is indeed important to be able to identify the deposits which remain visible long after their appearance. otherwise we run the risk

of distorting the statistics. The difficulty is to have a set of satellite data with adequate revisit time and an identical observation configuration from one scene to another. This problem also exists for avalanche deposit detection using SAR imaging but with a few days of revisit time it is possible to investigate its effects. For SPOT6-7, what would be the best strategy?
* * *

---

## Author Comment (AC2) · 13 Aug 2019

Dear anonymous reviewer

Thank you very much for your review of our paper. Please find in the following our answers to your general and specific comments:

1- Regarding the overall organization of the paper:

As you correctly state we give a lot of information on different satellite sensors we did test but then not use for the final mapping. In our opinion this part is necessary for the paper as it defines the narrative how we get to the decision to select SPOT6/7

data. Furthermore, we are convinced that the findings we describe in chapter 2.3 are very helpful for everyone considering to map an avalanche cycle with satellite data in the future. Therefore, we want to keep these sections. But to reduce the imbalance you state we will expand the information on the mapping procedure in the revised manuscript as stated in the answers to reviewer #1.

2- Figures:

We will improve the figures in the revised manuscript adding additional information such as contour lines.

3- Replication of the mapping elsewhere:

We were very lucky both in 2018 and in 2019 to have good acquisition weather just after the avalanche period and the financial and organizational means to order imagery available thanks to the Federal Office for the Environment and the National Point of Contact NPOC of swisstopo. Generally speaking the limits certainly are cloud cover, satellite tasking availability and costs. However, the availability of satellites for specific tasking is better in winter and if necessary other satellites with similar image characteristics might also be used (as described in chapter 2.2). Besides those two factors, the method we described is replicable anywhere. Depending on the resources for pre-processing and the experience of the person manually mapping the avalanches, it may consume more or less time. Like mentioned in chapter 4.3 it took us approximately 600 hours to map an area of 12'500 km2.

4- data availability:

As mentioned in the section "Data availability" toward the end of our paper the data is already available on request on ENVIDAT (https://www.envidat.ch/ui/#/metadata/spot6-avalanche-outlines-24-january-2018)

5- Mapping methodology:

As stated in the paper the mapping was done entirely manually. The NIR band was

applied to visualize the satellite data as base for the mapping (chapter 3). We do not understand how the NIR band could further help to classify avalanche zones automatically.

6- Mapped avalanches:

We are now conducting additional statistics mentioned but, in our opinion, presenting those additional statistics would go beyond the scope of this paper, which focus on the applied datasets and the mapping methodology. We plan to present the complete statistics of the mapped avalanches and an in-depth discussion in a follow up publication.

7- Large scale hazard mapping:

At the moment we are not able to calculate the avalanche terrain for entire Switzerland. We are now computing different smaller regions in Switzerland, Italy and further abroad after the method of Bühler et al. (2018). So as for now, we cannot apply our avalanche terrain mask but plan to do so in the future.

8- terrain types:

We do not completely understand this point. What types of forests are meant here? What water areas? Lakes maybe but most alpine lakes are frozen and their surface can be treated as flat terrain.

9- Age of avalanches:

Since the occurrence of large avalanche periods cannot be predicted well in advance, we believe it is not possible to plan on distinguishing the age of single avalanches based on two subsequent datasets. Even if theoretically possible, the costs for the tasking and acquisition of a pre-event scene would be too high. Depending on the area and available data, supplementary information like observations and photographs from local people might be used to identify pre-existing avalanches for the statistics. If this is not possible the results of our examination could be an indicator when doing statistics

even though the exact percentage will remain unknown. The outlines itself are relevant despite the age since the release time is not important in hazard zone mapping, for the evaluation of protection measures or for the validation and further development of numerical avalanche simulation software.

References:

Bühler, Y., von Rickenbach, D., Stoffel, A., Margreth, S., Stoffel, L., and Christen, M.: Automated snow avalanche release area delineation – validation of existing algorithms and proposition of a new object-based approach for large-scale hazard indication mapping, Nat. Hazards Earth Syst. Sci., 18, 3235-3251, 10.5194/nhess-18-3235-2018, 2018.

---

## Author Response (AR1)

Point-by-point changes applied to the Manuscript tc-2019-119

In this document we describe the applied changes. The detailed argumentation why we do so is given in the answer to the reviewers published in the open discussion.

Changes based on the comments of the reviewers:

Title: I believe that the title does not entirely represent the work that you have done. Data acquisition was rapid; however, mapping took 600 hours. Is 'large' the correct term when talking about a period of time? And in the end, you are only using Spot6/7 data, so maybe you could simply say so. Maybe the title could be something like: 'Mapping of an extreme avalanche cycle in the Swiss Alps using Spot6/7 optical images'.

We changed the title into "Where are the avalanches? Rapid SPOT6 satellite data acquisition to map an extreme avalanche period over the Swiss Alps"

2 Avalanche periods and data acquisition 2.1 Avalanche periods 2018 There are a lot of Swiss place names in this section. I – and probably the majority of readers – are not familiar with where all those places are. You could either provide a map, or leave this place names out.

We deleted all unnecessary names and added the most important names we wanted to keep to maps in the Figures 2 and 4

2.2 Rapid satellite data acquisition period I (8-10 January 2018) As mentioned above, I am a little bit unsure if all the detailed information you are providing here is necessary to get the full story of your paper. You are introducing a lot of data in a table and a figure, however, only a very tiny fraction of it is actually used for age-tracking of detected avalanches. At the same time, this section and also 2.3 read like discussion with some valuable information about which datasets to use for mapping purposes. So I am wondering if 2.2 and 2.3 could be deleted and some of the information could go into the discussion instead?

We deleted Figure 2 showing the location of the acquired imagery. But we wanted to keep table one to provide the readers with an overview of the data. We shortened the chapters 2.2 and 2.3. The detailed argumentation is given in the document with the answers to the reviewer.

2.4 Rapid satellite data acquisition period II (21 – 23 January 2018) Figure 2 caption Study area in the caption, however research area in the legend. Also, maybe write 'danger level 5'.

We changed the name to study area throughout the document consistently.

3 Mapping methodology You used additional information for the mapping. How were these data layers used? Did you mask out areas that were not considered avalanche terrain or did you blend these layers in and out?

We expanded the sentence to "The mapping itself was conducted manually by one person using a scale of 1:5 000. Additional data was blended in and out or swiped."

The flow direction check, was that done also by visual interpretation or with a GIS tool? The rechecking of accuracy of outlines and completeness. Could you give more details here as to what is meant by accuracy and what completeness means? Could you explain in some sentences if one or

more experts visually interpreted the images? If the rechecking was done by the same person as the mapping? I think this is operational implications as to whether the person mapping was an expert in remote sensing image interpretation or if this is a task that a less experienced person with avalanche background could also do? What is meant by feasible attributes that were described and which image examples were added when appropriate? These statements are not entirely clear to me.

We extended the sentence as following: "After the initial mapping, the outlines were checked twice, first using the 1:25 000 map to visually check flow direction and second the satellite imagery for rechecking the outlines and their completeness with different display parameters optimized once for areas in cast shadow and once brightly illuminated regions.

In order to keep all information that can be extracted from the images besides the outlines we previously defined an example key. It serves as a guide on how to record the metadata for each avalanche outline mapped with the presented method. In addition to a verbal description it includes illustrated examples wherever possible."

Figure 3: I am very interested in which attributes were assigned to these avalanches according to Table 2. What is the quality outline of these avalanches and could you maybe indicate by different line coloring where it is exact, estimated and created? I am in particular also interested in where the start and deposit zone altitude are and the fracture width. In example 2), the digitized avalanches show only parts of the slide path and the depositional area. Why are you not showing the start zone? How sure are you of the interpretation of the two avalanches in the shadow and is there maybe a third avalanche in the path to the left? Can you explain a little bit more in detail how the flow direction interpretation works when dealing with overlapping avalanches and indicate it in the figure?

We added the requested information to Figure 2 and we have chosen an example for 2b showing complete outlines. How we have handled flow direction was already described earlier.

Figure 4 and related text: I am curious about in which regions high avalanche activity was suspected. You are also talking about the effects of a high snowline in two valleys; however, I don't know where these valleys exactly are. Could you provide more topo- graphic information in Figure 4 for all non-Swiss readers? I wonder if a shaded relief as background would improve this figure too. Finally, the examples of avalanche density per km2 appear to be transparent. These small squares are very hard to see. Could you please improve the image quality or make these squares a little bit larger? In these examples you are going from 5 to 75 % coverage in 10 % steps. I wonder if you could do the same in the map. You would end up with fewer classes which would maybe increase the contrast between classes. Or how about using the AAI from Schweizer 2003 to represent avalanche activity?

We added important topographic information to the Figures 2 and 4. We experimented with making the squares larger and using a shaded relief but did not find it a cartographically satisfying option. So we propose to use an entire page to display this figure. Doing so makes it possible to clearly read all the details.

Figures 5 and 6: This is just a minor issue; however, these two figures could be visually slightly more pleasing. Related to Figure 6, I would find it very interesting to see examples of the different avalanche types and avalanche sizes. If it is too much for this article, maybe in an appendix figure.

We adapted the visualization of the Figures 4 and 5.

Figure 7: The largest of the blue outlines seems to go uphill (from the left upper corner towards the middle of the image) and then downhill again. Could you indicate flow directions maybe? These images are very hard to interpret as it is not really possible to understand where up and down is. Maybe contours or other topographical information would help here. Are these the smoothed lines or the raw digitized ones?

We added transparency to the imagery so that the topographic map shines through depicting the contour lines for better interpretability.

4.3 Potential improvements and follow up analysis You are talking about using machine learning for automatic analysis. Could you expend on these thoughts a little bit more and explain how such a workflow might look like and which kind of algorithms would be suitable here. You mention that a lot of background knowledge and map interpretation skills are involved in mapping. Is there a way that machine learning could incorporate these skills too? And how confident are you that a ML algorithm could classify avalanche types?

We expanded this section "Therefore it would save a lot of time and costs if the data could be analyzed automatically for example by machine learning (Zhang et al., 2016) which was already successfully applied to detect landslides. An overview on potential algorithms and workflows is given by Ghorbanzadeh et al. (2019)."

4. In your aim of this paper you are stating that a nearly complete database of avalanche activity could be used to validate the avalanche bulletin. This is, however, missing, and could easily be added in the discussion.

We added the following section to the discussion:

**4.4 Validation approaches for avalanche bulletins**

The avalanches mapped with the method described in this paper might be used for a validation of the avalanche bulletin. The European avalanche danger scale defines that with very high avalanche danger (level 5) "numerous very large and often extremely large natural avalanches can be expected" (https://www.avalanches.org/standards/avalanche-dangerscale/). No additional definition of the terms "numerous" and "often" nor the time period or area this would apply to is given. Defining those two terms we can check the density for the very large (size 4) and extremely large avalanches (size 5) for each avalanche warning region in the study area for the time period mapped. With the two calculated densities we will be able to determine where the danger level forecasted in the bulletin was correct, too high or too low. (Because of the definitions of the EAWS danger scale this kind of evaluation is only applicable for high or very high avalanche danger (level 4 and 5).) Of course, the uncertainties and limitations mentioned in other sections of this paper need to be considered for that. Additionally, the mapped avalanches allow for a verification of the forecasted release heights and aspects. With this information we can provide invaluable feedback to the avalanche warning service and help them improve their validation and forecast in future situations.

5. This is a case study from Switzer- land. Since satellites cover pretty much all snow-covered mountain areas worldwide, it would be interesting to read a short discussion on the potential of doing this anywhere else. Here data availability, revisit time, max spatial coverage, cloud cover and shadow problematic and also acquisition costs could be discussed.

We added the following section to the discussion:

[revised manuscript text omitted]

07 January 2018, 10:24                           | 0.3 PAN
1.2 Multispectral | PAN: 450 - 800
Red: 655 - 690
Green: 510 - 580
Blue: 450 - 510
Near-IR: 780 - 920                 | 107
107
Total: 214           |
| Pléiades (optical)    | 06 January 2018, 11:07                                                     | 0.5 PAN
2.0 Multispectral | PAN: 480-830 nm
Blue: 430-550 nm
Green: 490-610 nm
Red: 600-720 nm
Near-IR: 750-950 nm            | 130
143
Total: 237           |
| SPOT6/7 (optical)     | 06 January 2018, 09:52
06 January 2018, 10:58                           | 1.5 PAN
6.0 Multispectral | Blue: (455 nm – 525 nm)
Green: (530 nm – 590 nm)
Red: 625 nm – 695 nm)
Near-IR: (760 nm – 890
nm) | 265
804
1751
Total: 2820  |
| TerraSAR-X (radar)    | 06 January 2018, 17:00
08 January 2018, 05:44
09 January 2018, 05:27 | 1 SpotLight
3 StripMap    | X-band, 8 – 12,4 Ghz                                                                                          | 2 * 116
2 * 1500
Total: 3232 |

**Table 1: Acquired satellite datasets for the first period.**

[revised manuscript text omitted]

**3 Mapping methodology**

We used visual interpretation to identify and digitize avalanches as polygons over the whole study area. To improve visibility in both, illuminated and shaded areas, we modified contrast and brightness, used 25 image stretching and gamma optimization. Since the optimal brightness and contrast vary for different

|-------------|---------|--|--|
|             |         |  |  |
| Gelöscht: F | igure 2 |  |  |
|             |         |  |  |
|             |         |  |  |
|             |         |  |  |
